# CONVOLUTIONAL NEURAL NETWORKS ARE NOT INVARIANT TO TRANSLATION, BUT THEY CAN LEARN TO BE

## ABSTRACT

When seeing a new object, humans can immediately recognize it across different retinal locations: we say that the internal object representation is invariant to translation. It is commonly believed that Convolutional Neural Networks (CNNs) are architecturally invariant to translation thanks to the convolution and/or pooling operations they are endowed with. In fact, several works have found that these networks systematically fail to recognise new objects on untrained locations. In this work we show how, even though CNNs are not 'architecturally invariant' to translation, they can indeed 'learn' to be invariant to translation. We verified that this can be achieved by pretraining on ImageNet, and we found that it is also possible with much simpler datasets in which the items are fully translated across the input canvas. Significantly, simply training everywhere on the canvas was not enough. We investigated how this pretraining affected the internal network representations, finding that the invariance was almost always acquired, even though it was some times disrupted by further training due to catastrophic forgetting/interference. These experiments show how pretraining a network on an environment with the right 'latent' characteristics (a more naturalistic environment) can result in the network learning deep perceptual rules which would dramatically improve subsequent generalization.

## 1 INTRODUCTION

The equivalence of an objects across different viewpoints is considered a fundamental capacity of human vision recognition (Hummel, 2002). This is mediated by the inferior temporal cortex, which appears to provide the bases for scale, translation, and rotation invariance (Tanaka, 1996; O'Reilly & Munakata, 2019). Taking inspiration from biological models (LeCun et al., 1998), Artificial Neural Networks have been endowed with convolution and pooling operations (LeCun et al., 1998; 1990). It is often claimed that Convolutional Neural Networks (CNNs) are less susceptible to irrelevant sources of variation such as image translation, scaling, and other small deformations (Gens & Domingos, 2014; Xu et al., 2014; LeCun & Bengio, 1995; Fukushima, 1980). While it is difficult to overstate the importance of convolution and pooling operations in deep learning, their ability to make a network invariant to image transformations has been overestimated: for example, Gong et al. (2014) showed that CNNs achieve neither rotation nor scale invariance. Similarly, multiple studies have reported highly limited translation invariance (Kauderer-Abrams, 2017; Gong et al., 2014; Azulay & Weiss, 2019; Chen et al., 2017; Blything et al., 2020).

It is important to understand the reason for the misconception regarding the ability of CNNs to be invariant to translation. We believe this is due to two misunderstanding. Firstly, it is commonly assumed that CNNs are 'architecturally' invariant to translation (that is, the invariance is built in the architecture through pooling and/or convolution). For example: "[CNNs] have an architecture hard-wired for some translation-invariance while they rely heavily on learning through extensive data or data augmentation for invariance to other transformations" (Han et al., 2020), and "Most deep learning networks make heavy use of a technique called convolution (LeCun, 1989), which constrains the neural connections in the network such that they innately capture a property known as translational invariance. This is essentially the idea that an object can slide around an image while maintaining its identity; a circle in the top left can be presumed, even absent direct experience) to be

the same as a circle in the bottom right." (Marcus, 2018), see also LeCun & Bengio (1995); Gens & Domingos (2014); Xu et al. (2014); Marcos et al. (2016).

In fact, the convolution operation is translationally equivariant, not invariant, meaning that a transformation applied to the input is transferred to the output (Lenc & Vedaldi, 2019). Even when this point is made, such in LeCun & Bengio (1995), is it still assumed that the equivariance is enough to support an important degree of translation invariance. For example, LeCun & Bengio (1995) write: "Once a feature has been detected its exact location becomes less important as long as its approximate position relative to other features is preserved". As a matter of fact, equivariance and invariance are mutually exclusive functions (a representation cannot support both), and accordingly, any invariance supported by a network must be coded into the fully connected part rather than in the equivariant convolutional layers. Moreover, perfect equivariance can be lost in the convolutional layers (Azulay & Weiss, 2019; Zhang, 2019) through subsequent sub-sampling (implemented with pooling and striding operations, commonly used in almost any CNN). Therefore, overall, most modern CNNs are neither architecturally invariant nor perfectly equivariant to translation.

The other reason for overestimating the extent that CNNs are invariant to translation resides in the failure to distinguish between trained and online translation invariance (see Bowers et al. 2016). Trained invariance refers to the ability to correctly classify unseen instances of a class in trained location. For instance, a network trained on the whole visual field on identifying instances of dogs, will be able to identify a new image of a dog across multiple locations. This feature is obtained by data-augmentation: jittering the training samples so that the network is trained on items across different locations (Kauderer-Abrams, 2017; Furukawa, 2017). However, this should not be considered a form of translation invariance, as it is simply a case of identifying a test image (a novel image of a dog) at a trained location. More interesting is the concept of 'online' translation invariance: learning to identify an object at one location immediately affords the capacity to identify that object at multiple other[1].

Online translation invariance is generally measured by training a network on images placed on a certain location (generally the center of a canvas), and then testing with the same images placed on untrained location. In many reports CNNs performed at chance level on untrained locations (Kauderer-Abrams, 2017; Gong et al., 2014; Azulay & Weiss, 2019; Chen et al., 2017; Blything et al., 2020). This problem has been tackled with several architectural changes: Sundaramoorthi & Wang (2019) suggested a solution based on Gaussian-Hermite basis; Bruna & Mallat (2012) used a wavelet scattering network model; Jaderberg et al. (2015) added a new module that can account for any affine transformation; Blything et al. (2020) used Global Average Pooling.

Without having to resort to new architectures, two works have recently contrasted the previous findings, obtaining a high degree of online translation invariance: Han et al. (2020) found that a CNN exhibited an almost perfect online translation invariance on a Korean Characters recognition task, but it did not compare these results with the previous literature. Blything et al. (2020) also found perfect online translation invariance when a VGG16 network was pretrained on ImageNet but found almost no invariance when the same (vanilla) network was untrained. This latter finding explains Han et al. results, as they also used a pretrained network when assessing online translation invariance. Together these results hint to the fact that translationally invariant representations do not need to be built inside the network architecture, but can be learned. In the current work, we further explore this idea.

## 2  CURRENT WORK

In this work we focus on 'online' translation invariance on a classic CNN, using VGG16 (Simonyan & Zisserman, 2014) as a typical convolutional network. We show how, even though classic CNNs are not 'architectural' invariance, they can 'learn' to be invariant to translation by extracting latent features of their visual environmenta (the dataset). Learning is used in the sense that the invariance is coded within the network weights, optimized through backpropagation, and not hard-wired in the network architecture, such as Sundaramoorthi & Wang (2019) or Bruna & Mallat (2012).

---

[1] 'Trained translation invariance' in which images can be identified across the canvas after training exemplar images across many locations is not to be confused with our approach of 'training' translation invariance, in which a network is trained to exhibit 'online' translation invariance

We trained on environments in which the key characteristic was that items' categories were independent on their position (for CNNs learning the categories based on their position, see Semih Kayhan & van Gemert 2020). Our main contribution is finding that by pretraining on such environments, CNNs would indeed learn to be invariant to translation.

Why is this important? First, it is important to know how CNNs work, and there is currently confusion about how and when CNNs support translation invariance. Second, a network that learns deep characteristic of its visual environment such as being invariant to translation, rotation, etc., is able to accelerate subsequent training, and accordingly, it is important to understand the conditions that foster invariance. In addition, because CNNs have been recently suggested as a model for the human brain (Richards et al., 2019; Ma & Peters, 2020; Kriegeskorte, 2015; Zhuang et al., 2020), it is important to understand if and how they can learn fundamental perceptual properties of human vision, of which invariance to translation is one (Blything et al., 2020; Bowers et al., 2016; Koffka, 2013).

## 3 OVERVIEW OF THE EXPERIMENTS

Blything et al. (2020) found that using a VGG16 network pretrained on ImageNet would result in an almost perfect translation invariance on a different dataset in which items were trained only on one location. They compared these results with a vanilla network, that is a non-pretrained network, which showed a lack of translation invariance. Here we replicate these findings with a wider variety of datasets (Section 3.2). We then show that it is possible to obtain similar results using much simpler artificial datasets in which objects were fully-translated across the canvas, but with some limitations due to the difference between the pretraining and the fine-tuning datasets (Section 3.3). As a sanity check, we showed that pretraining on the whole canvas is not enough to acquire translation invariance, but the network must be pretrained on fully-translated objects (Section 3.4). Next, we report studies that assess translation invariance learned from partial information in the environment, that is, whether invariance can be generalized to the whole visual field when the network was only trained on one area of the visual field, and whether invariance extends to all classes when only a subset of classes was trained at all locations (Section 3.5). We found that generalization failed in these cases. Finally, we report a cosine similarity analysis that provides some insight into the learned internal network representations (Section 3.6), and shows that the poor performance on some conditions in Section 3.3 was likely due to catastrophic forgetting/interference.

### 3.1 DATASETS

We used six datasets spanning a high range of complexity. From the more complex to the less complex[2], we used: EMNIST (Cohen et al., 2017); FashionMNIST (FMNIST), from Xiao et al. (2017), Kuzushiji MNIST (KMNIST), from Clanuwat et al. (2018); MNIST, from (LeCun et al., 1998); and two versions of the Leek dataset used in Blything et al.: one contained only 10 images instead of 24 of the original dataset (Leek10). The other contained only two images from the original dataset (Leek2), disjointed from the images used for Leek10. Representative examples (and, for Leek10 and Leek2, the entire datasets) are shown in Figure 1B. We did not apply any data-augmentation to the datasets (apart translating the items on the canvas, as explained below).

### 3.2 EXPERIMENT 1: PRETRAINING ON IMAGENET

The experimental design is shown in Figure 1A: we tested a VGG16 network either pretrained on ImageNet or not pretrained (vanilla). Both networks were then re-trained on a 1-location dataset, that is, a datasets where items from all classes were presented only on one location. The network that was pretrained on ImageNet would therefore use the learned parameters as initialization for the new training with the 1-location dataset (this is commonly referred as fine-tuning, Girshick et al. 2014), whereas the vanilla network would start from scratch with Kaiming Initialization (He et al., 2016). We used Adam optimizer with a fixed learning rate of 0,001. The 1-location dataset used items from

---

[2]For EMNIST, FashionMNIST, KMNIST and MNIST we based our judgment of complexity on the benchamrk in `https://paperswithcode.com/task/image-classification`. We deemed Leek10 and Leek2 as the easiest to classify due to the lack of intra-class variability, as each class is composed of just one image

the six datasets described in Section 3.1, but with each item resized to $50 \times 50$ pixels and placed on the leftmost-centered location on a black canvas $224 \times 224$. Therefore no translation was used for this dataset. Both networks were then tested on the same items placed across the whole canvas (we did not place the items at the edge of the canvas to avoid cropping). Therefore, the networks were tested on their ability to recognise trained objects on unseen locations ('online' translation invariance). We repeated each condition 5 times.

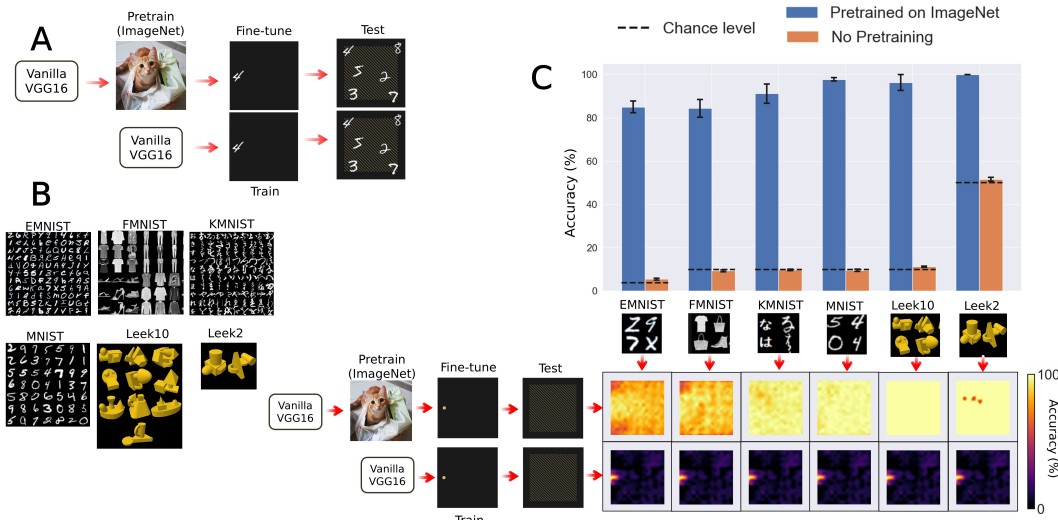

Figure 1: A. Representation of the general experimental design used here and in following experiments. B. Samples of the six datasets used here. Leek10 and Leek2 are shown in their entirety. C. Results from experiment 1. The barplot indicates the average accuracy across the 5 runs. Dashed lines indicates chance, and error lines represents one standard deviation. The heatmaps at the bottom show the average accuracy across all classes for each dataset at each location in the canvas (interpolated across $19 \times 19$ test points used).

The results are shown in Figure 1C. A network pretrained on ImageNet was able to recognize with high accuracy objects from the 1-location datasets when tested on unseen locations, whereas a vanilla network was not able to do so (Figure 1C, barplot). For the pretrained condition, accuracy seemed to slightly increase with simpler datasets. We can better understand the extent of networks' invariance to translation by plotting its accuracy across the whole canvas: we tested the network on items centered across a grid of $19 \times 19$ points equally distributed upon the canvas. By interpolating the results across the tested area we obtained a heatmap like those shown in Figure1C, bottom. We can see that in the case of the vanilla network (bottom row), for all datasets, high accuracy was achieved only at trained location (the leftmost-centered one). In fact, performance started degrading after a displacement of around 10 pixels, and was at chance at 30 pixels. On the other hand, the pretrained model (upper row) was extremely accurate everywhere on the canvas.

### 3.3 EXPERIMENT 2: BEYOND IMAGENET

We hypothesised that the network pretrained on ImageNet learned translation invariance because of the data augmentation performed during pretraining that involved translating subsets of the input images across the visual field of the network. If this is correct, we could hypothetically make a network learn to be invariant to translation with any simple datasets in which the items occur all across the canvas. There are obviously other features that a network can extract from ImageNet that are not related to image translation, but here we will focus on translation only.

We pretrained a VGG16 network on fully-translated datasets: items from datasets in Section 3.1 were resized to $50 \times 50$ pixels and randomly placed anywhere across a $224 \times 224$ black canvas. Similarly to the previous experiment, we then fine-tuned the network on each 1-location dataset, and tested it on the same items, fully-translated. The experimental setup is illustrated in Figure 2A. Again, if the network has learned to be invariant to translation, it would be able to recognise objects

everywhere on the canvas, without need to be trained on every location. The resulting accuracy, averaged across the whole canvas and across five repetitions for each (pretrain, fine-tune) pair, are shown in Figure 2B. Notice that the accuracy are normalized across datasets so that 0 is chance level. In most cases, the networks were able to correctly classify items from the 1-location datasets when seen in untrained locations (unlike in Figure 1C where we saw that a vanilla network was not able to perform above chance in untrained locations). However, this is not true for all combination of pretrained and fine-tuned datasets. We observed a pattern in which networks pretrained with a "complex" fully-translated dataset obtained high performances when fine-tuned on a 1-location "simple" datasets, but the opposite was not true (for example, pretraining on Leek2 resulted in poor performance when fine-tuned on any other dataset). By probing the networks' internal representation, we show in Section 3.6 how even networks pretrained on simple datasets had acquired translation invariance, but it was then disrupted by fine-tuning. Before that, we want to confirm that networks that acquired translation invariance did so because they experienced fully translated items, and not simply because they had been trained across the whole canvas.

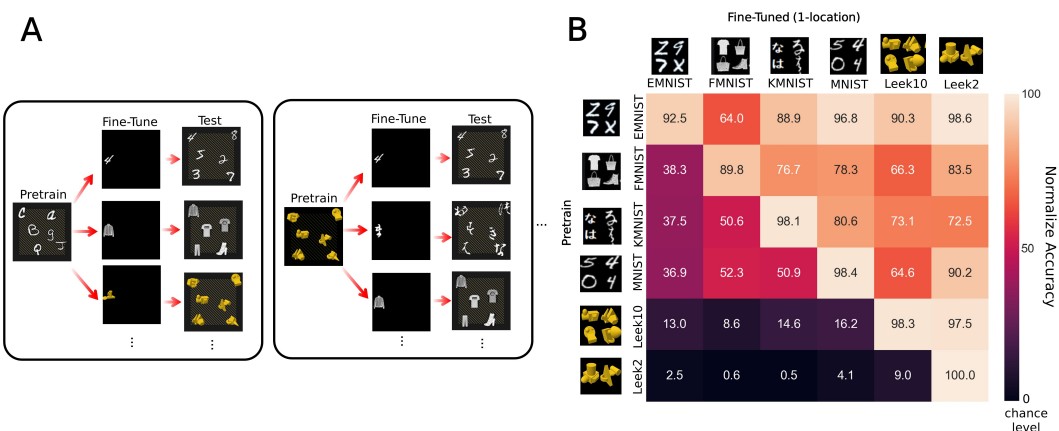

Figure 2: A. Experimental design for Experiment 2. B. Results in terms of normalized accuracy (0 is random chance, 100 is perfect performance). Standard deviation for each condition is shown in Section A.1

## 3.4 EXPERIMENT 3: TRAINING ON THE WHOLE CANVAS IS NOT ENOUGH

It is conceivable that in the previous experiments, and in similarly designed experiments in the literature (Kauderer-Abrams, 2017; Gong et al., 2014; Chen et al., 2017; Blything et al., 2020), vanilla networks failed to show invariance to translation because they were tested on locations where they had not seen any items. In which case, pretrained networks succeeded not because they had acquired the deep property of translation invariance from the visual environment, but simply because they had been trained on the whole canvas. To test this hypothesis we separated the canvas in 9 equilateral areas ($58 \times 58$ pixels), and within each area, only 2 of the total classes were presented. The

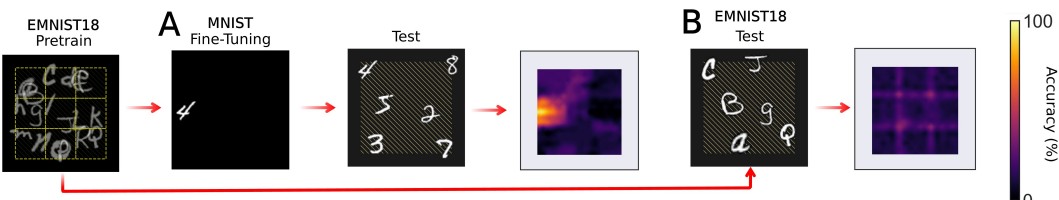

Figure 3: Experimental design for Experiment 3 and results. A. After fine-tuning on MNIST, the network did not generalize on untrained locations. B. When tested on the same dataset (EMNIST18), but without using classes segregation, letters were only recognised when presented on the area they were trained on, so mean accuracy was low. In both cases, the heatmaps are averaged across all classes.

items were randomly centered anywhere within their area (in such a way that part of the object could sometime slightly overlap another area, but the objects were never cropped). Therefore the objects were subjected to limited translation. Once trained with this setup, we fine-tuned the network on a 1-location dataset and tested the same dataset on the whole canvas (like in Section 3.2 and 3.3). We used the EMNIST dataset with the first 18 categories (EMNIST18) for pretraining, and MNIST for 1-location fine-tuning and fully-translated testing dataset, because they were the datasets that most consistently resulted in good translation invariance in the previous experiment. Results are shown in Figure 3A. We also tested the pretrained network on a fully-translated version of EMNIST18, that is, without class segregation, and without fine-tuning (Figure 3B). Even though the network was trained on items everywhere on the canvas, it did not acquire the ability to generalize on unseen locations with neither EMNIST18 nor newly trained objects (MNIST). This is a strong demonstration that the network needs to be trained on an environment where objects are translated across the whole canvas in order to learn to be invariant to translation, and that simply training on the whole canvas is not enough.

### 3.5 EXPERIMENT 4: TRAINING ON LIMITED TRANSLATIONS IS NOT ENOUGH

In the previous experiments we assessed the degree of online translation invariance that a network exhibited with items trained at one location. We showed that a network would display translation invariance only if it had been pretrained on a fully-translated dataset. Here we explore alternative visual environments that may allow a network to learn this fundamental principle.

In the following conditions, we used characters from EMNIST, and tested on a fully-translated EMNIST dataset. Whereas in the previous experiments we fine-tuned on a new dataset, here we are interested in generalizing on the same items the network is trained on, so we do not use fine-tuning (however, testing on a fine-tuned dataset is examined in Section A.2). We divided the canvas in 4 quadrants and treated the upper-right quadrant in a special way in the following three conditions.

In **Condition 1** the upper-right quadrant was left empty, and all the other classes were trained on the rest on the canvas. When tested, the network was only partially able to generalize on the untrained area (Figure 4, left).

In Conditions 2 and 3 we explored the possibility that training different classes on different areas would allow the network to recognize the classes outside their training area. Unlike the experiments in Section 3.3, in which new classes were trained at one location *after* the fully-translated pretraining had occurred, here we train, at the same time, fully-translated classes and partially translated classes.

In **Condition 2** we trained the letter T to Z of the EMNIST datasets on the upper-right quadrant, and the remaining letters on the rest of the canvas. When tested on a fully-translated version of that dataset (in which there was no class segregation) the network failed: the letters would be recognized only in the respective areas (Figure 4, right).

In **Condition 3** we slightly modified the previous experiment by training a subset of classes across the whole canvas, while limiting another subset to the upper-right quadrant. We split the experiment in 4 sub-conditions, varying the number of fully-translated classes and limited translated ones. In all sub-conditions the network did not achieve full translation invariance on classes that were presented only on the upper-right quadrant (more details in Section A.2.3 and Figure 9).

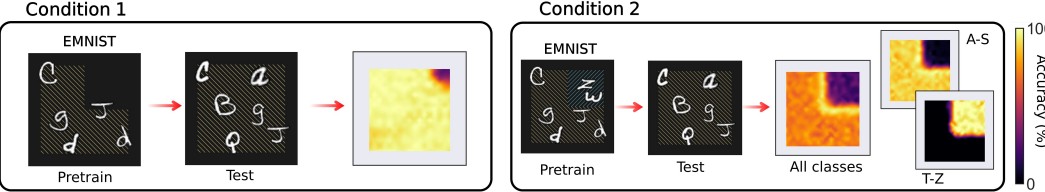

Figure 4: Experimental design and results of condition 1 and 2 in Experiment 4. On Condition 1, the heatmap shows a certain degree of generalization on untrained area, but not complete. On Condition 2, the network only accurately predicted classes on the area in which they were present during training.

The overall conclusion from these experiments is that, in order to acquire the property of being invariant to translation, a network needs to be pretrained on fully-translated objects, for all classes. Further details on these experiments, including the results when fine-tuned on a new dataset, are provided in Section A.2.

### 3.6 EXPERIMENT 5: INTERNAL REPRESENTATION WITH COSINE SIMILARITY ANALYSIS

We found in Experiment 2 in Section 3.3 that CNNs would some times not display invariance to translation after fine-tuning. This seemed to happen when they had been pretrained on a simple dataset and fine-tuned on a complex one, and it could be explained by the phenomenon of inter-ference and catastrophic forgetting (Furlanello et al., 2016; McCloskey & Cohen, 1989) in which training on new tasks degrades previously acquired capabilities. If this was true, it would mean that a network could learn to be invariant to translation *with any translated dataset*, regardless of its complexity, and this ability could be retained with techniques that prevent catastrophic forgetting (Beaulieu et al. 2020; Javed & White 2019; Li & Hoiem 2016).

In order to test the hypothesis that catastrophic interference may play a role, we computed the cosine similarity between the activations of the penultimate networks' layers when given as input an image at the leftmost-centered location and an images at different levels of displacement from that location. This metric indicates the degree to which translated objects share the same internal representation (we show the results of applying the cosine similarity analysis across the whole network in Section A.3.2).

We firstly checked that the cosine similarity analysis correctly captured the increasing similarity of internal representation with translated images for a vanilla network, a network pretrained on Ima-geNet, and a network pretrained on the tested dataset (we used the networks trained in Sections 3.2 and 3.3). Results are shown in Figure 5A. Notice that the vanilla and pretrained models were never trained on the items used to compute the cosine similarity. For the vanilla network, the similarity quickly dropped, indicating that more translated objects have very different internal representations. The other two networks have a very high similarity even for highly translated objects. This is an-other confirmation that networks trained in such environments did develop a general ability of being invariant to translation.

We then measured the amount of catastrophic forgetting by computing the cosine similarity of each pretrained network (on each dataset in Section 3.1) before and after fine-tuning on the 1-location datasets. Results for all conditions, at different experimental stages, are shown in Figure 5B. For most datasets, the cosine similarity was high for every dataset before fine-tuning, even for untrained

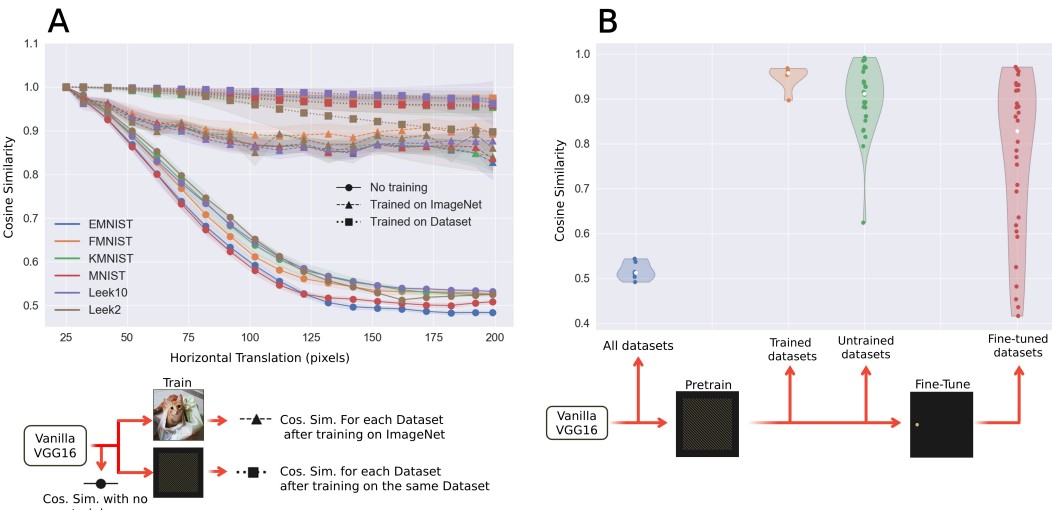

Figure 5: A. Cosine similarity across several horizontal displacements. B. Cosine similarity at different stages of the experimental setup for experiments in Section 3.3. The bottom part of the figure indicates the different stages at which the analysis is performed.

items. After fine-tuning, however, the cosine similarity would sometime go down to an un-trained level. In Figure 10 in Section A.3.1 we show that the change is consistent with the drop in accuracy we saw in Section 3.3.

# 4 DISCUSSION

Our experiments show that translation invariance can be learned in CNNs that lack built-in architectural invariance, and raise the possibility that a whole range of perceptual capabilities can be learned rather than built in the architecture. This work also suggest that, in certain cases, the architecture may be less important than the visual world the network is trained on. In fact, we verified that a simple fully-connected network (without convolutions) can indeed learn translation invariance in some conditions (see Section A.4 in the Appendix).

Neural networks performance is often compared to human performance (Baker et al., 2018; Han et al., 2019; Srivastava et al., 2019; Ma & Peters, 2020). A fundamental feature of human perception is that it supports widespread generalization, including combinatorial generalization (e.g., identifying and understanding images composed of novel combination of known features). Current CNNs are poor at generalizing to novel environments (Geirhos et al., 2018), especially when combinatorial generalization is required (Vankov & Bowers, 2020; Hummel, 2013). Here we have shown that CNNs are able to extract latent principles of translation invariance from their visual world and to re-use them to identify novel stimuli with very different visual forms in untrained locations. An interesting question is the extent to which other forms of generalization and other fundamental principles of perception (e.g., Gestalt principles of organization, Koffka 2013) can be learned in standard CNNs trained on the appropriate datasets, and what sorts of generalization requires architectural innovations. Without the right training environment, it is not surprising that CNNs fail to capture the cognitive capacity of the human visual system (Funke et al., 2020), and the only way to address this fundamental question is to train models under more realistic conditions.

Although training on more naturalistic datasets may lead to better and more human-like forms of generalization, it is also worth highlighting how our artificial hand-crafted environments, such as our fully-translated datasets, could be used to train networks to acquire a particular perceptual regularity in spite of the architecture not directly supporting that representation. This may prove to be a useful technique for learning in more complex environments: Instead of having a network learning all possible visual configurations (through data augmentation) of a given dataset, the network can be pretrained on extremely simple datasets that embed fundamental perceptual principles of the environment. When facing a complex dataset, the network only needs to learn the basic configuration of the new objects, and extrapolate the others through learned perceptual properties. This approach would clearly need to address the problem of catastrophic forgetting since, as we have seen, it can seriously impair generalization of learned perceptual rules when applied to new objects (Li & Hoiem, 2016; Furlanello et al., 2016). This approach would resemble Curriculum learning (Bengio et al., 2009), but instead of progressively learning new classes, the network would progressively learn new perceptual properties. We are not aware of any study adopting this approach.

# 5 CONCLUSION

We have shown that, even though standard CNNs are not architecturally invariant to translation, they can learn to be by training on a dataset that contains this regularity. We have also shown how such property is retained when fine-tuning on simpler dataset, but lost for more complex ones, and this appears to reflect the role of catastrophic interference in constraining translation invariance rather than a failure to learn invariance from these simple datasets. More generally, we suggest that by training artificial networks with non-naturalistic datasets, we are constraining their ability to learn deep fundamental rules of perception. By using more naturalistic environments, and an approach that would avoid catastrophic forgetting, we could unlock a greater capacity of generalization.

ACKNOWLEDGMENTS

We would like to thank Gaurav Malhotra for helpful comments and suggestions.

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

# A APPENDIX

## A.1 BEYOND IMAGENET

In Section 3.3 we pretrained on the fully-translated version of each one of the the six datasets described in Section 3.1. We then fine-tuned the network on each 1-location dataset, and tested it on the same items, fully-translated. We repeated each training session 5 times. In Figure 2 we show the average performance across each repetition. Here we also show the standard deviation for each one of the 36 conditions (Figure 6).

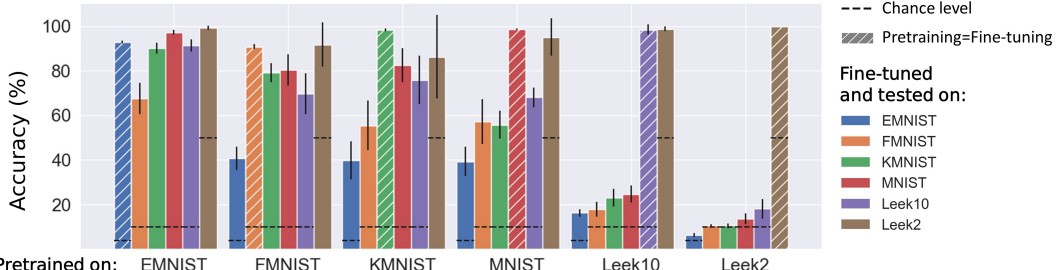

Figure 6: Average results across each repetition for Experiment 1. Error lines indicate one standard deviation. Hatches indicate the condition in which fine-tuning had the same items as pretraining (corresponding to the diagonal in Figure 2)

## A.2 TRAINING ON LIMITED TRANSLATIONS: FURTHER TESTS

### A.2.1 CONDITION 1: INTER-CLASS ANALYSIS

We showed that a CNN was only partially able to generalize on untrained locations (Section 3.5). However, we noticed a certain degree of inter-class variability: even though in most cases the results were similar to the average shown in Figure 3, for a few EMNIST letters the accuracy was high all across the canvas. We show in Figure 7 the accuracy across the canvas for each class, noticing that perfect translation invariance was achieved for letter E and Z, and several other classes reached an almost perfect invariance. It is not clear why there was this difference across classes. Notice that we could find a much lower degree of inter-class variability for the Condition 2, and that analysis is therefore not shown here.

### A.2.2 CONDITION 1 AND 2: FINE-TUNING

In Conditions 1 and 2 in Section 3.5 we trained a network with EMNIST items by translating them everywhere across the canvas apart from the upper-right quadrant. We then tested the network on the same dataset it was trained on (EMNIST), but fully translated. In this section, we show what happens with a experimental design similar to that used in Section 3.3, that is, by training on a new 1-location dataset and testing on this fully-translated dataset. In Condition 1, after having pretrained on the EMNIST dataset leaving empty the upper-right quadrant, we fine-tuned the network on 1-location MNIST and tested it on a fully-translated MNIST. As shown in Figure 8, left, we obtained very similar results to those in Figure 4, that is: the network showed translation invariance on locations all across the pretrained area, but only partial invariance on locations in which it had never been trained. We also analysed the inter-class variability finding that, like in Figure 7, the network showed perfect translation invariance for some of the classes in the MNIST dataset.

In Condition 2 in Section 3.5 we filled the upper-right quadrant with a subset of EMNIST classes, showing that the network learned to selectively recognise classes only on the trained location. In this section, we show what happened when fine-tuning the network on a new 1-location dataset. This is shown in Figure 8, right. We can see that the network had retained some kind of bias in that it would incorrectly classify items presented on the upper-right quadrant, but the separation was not as clear as before. As usual, we observed some inter-class variability regarding the network accuracy on the left-out quadrant, but in this case none of the classes reached perfect translation invariance.

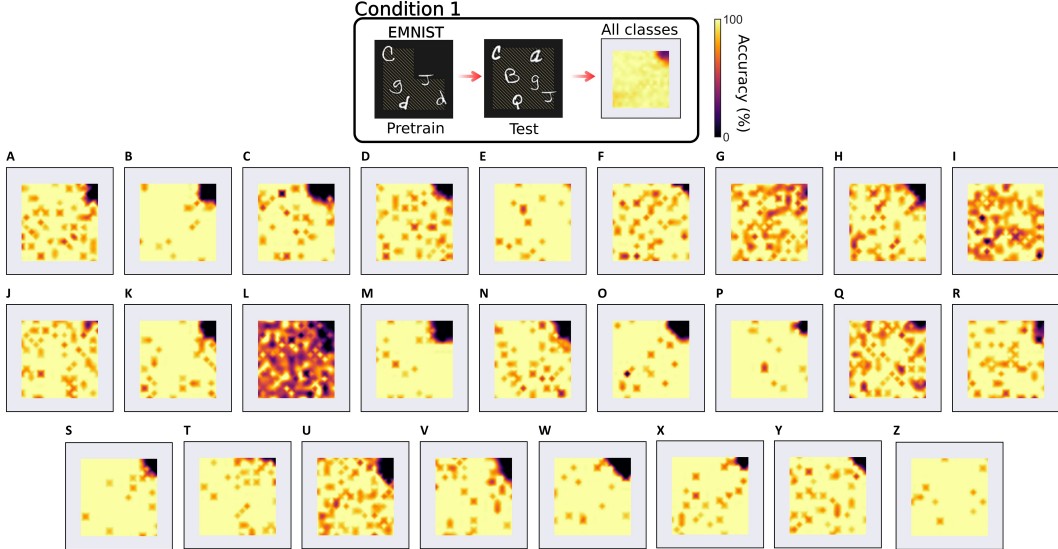

Figure 7: Heatmaps for each EMNIST class for the Condition 1 of Experiment 3. The network seem to be perfectly invariant to translation for some classes.

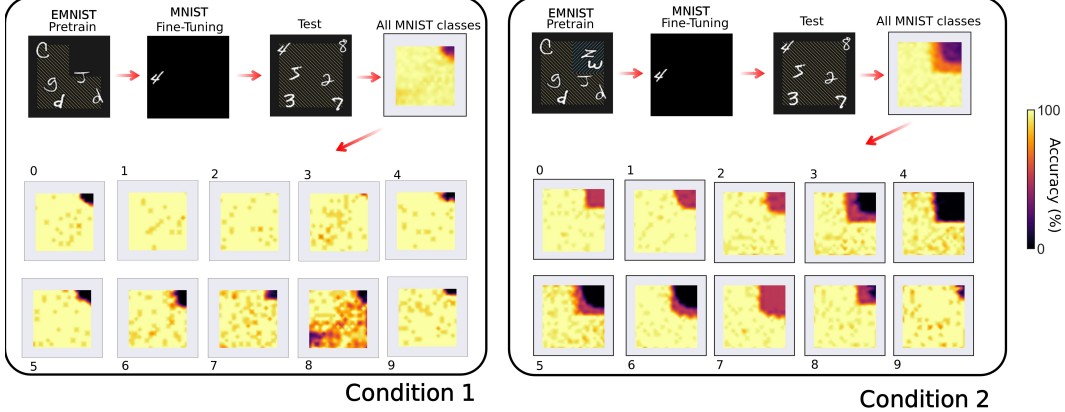

Figure 8: Condition 1 and 2 from Experiment 3. Here, instead of testing on the pretrain dataset, we fine-tuned on a new dataset (MNIST) and tested on the fully-translated version of it. We also show the heatmap for each class, showing a discrete degree of inter-class variability and perfect translation for some classes.

### A.2.3 CONDITION 3: MORE DETAILS

In this section we provided more details about the Condition 3 briefly described in Section 3.5. We used a dataset structured in this way: a subset of classes from EMNIST was placed anywhere on the canvas (fully-translated); the remaining classes were only placed on the right-upper quadrant. We varied the number of classes trained on the whole canvas in the following way: for Condition 3.1, only the class corresponding to the letter A was trained on the whole canvas; for Condition 3.2, the classes from A to J; for Condition 3.3, the classes from A to T; from Condition 3.4, the classes from A-Y. Therefore the amount of classes trained on the whole canvas, for each sub-condition, 1, 10, 20, and 25. We then tested the network on the same EMNIST dataset, but this time all classes were fully translated. The relevant measure here is the accuracy for the classes presented only on the right-upper quadrant, when then tested on the whole canvas, and in particular whether the network would be able to accurately classify those classes when presented on the area of the canvas in which they were not trained on. The results are shown in Figure 9. We analysed separately the results from the two classes. The interplay between these two groups (fully-translated and partially-translated) is

non-trivial, with the Condition 3.1 showing almost perfect translation invariance for the letter A and the condition 3.2 showing a slightly degraded performance for the fully-translated letters A-J on the right-upper quadrant. However, across all sub-conditions, it is clear that the letters trained on the right-upper quadrant were never recognized when presented elsewhere.

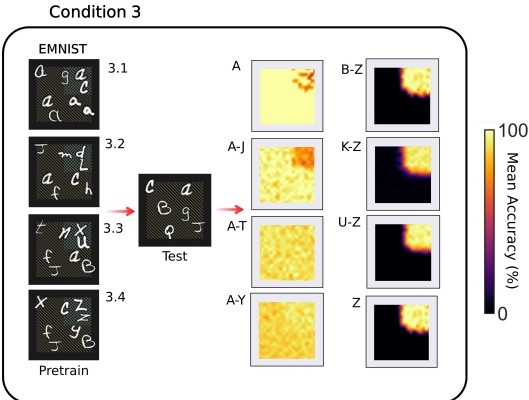

Figure 9: Condition 3 for Experiment 3. This condition was split in 4 sub-conditions, with increasing number of letters trained on the whole canvas. Heatmaps are shown for each group.

## A.3 COSINE SIMILARITY ANALYSIS

### A.3.1 BEFORE AND AFTER FINE-TUNING FOR EACH PRETRAINING DATASET

We showed in Section 3.6, Figure 5, the grouped results of the cosine similarity analysis before and after fine-tuning. In Figure 10 we show the results for networks pretrained on each datasets. These results are very consistent with the accuracy results shown in Section 3.3 and A.1: networks pretrained on complex dataset would mostly retain a high degree of cosine similarity after fine-tuning with most of the datasets, meaning that the translation invariance property was preserved. In other cases, and really strikingly with Leek10 and Leek2, the cosine similarity dropped after fine-tuning with all the other complex datasets. Significantly, a high degree of cosine similarity was observed before the fine-tuning, *for all datasets, and for networks pretrained on any dataset*. This implies that, for example, a network pretrained on a fully-translated Leek2 dataset (an extremely simple and fast-to-learn dataset) would acquire a certain degree of translation invariance even with more complex datasets, such as EMNIST. This property could be accessed if catastrophic forgetting were to be ameliorated during fine-tuning, or by using a method that would exploit the network internal representation itself.

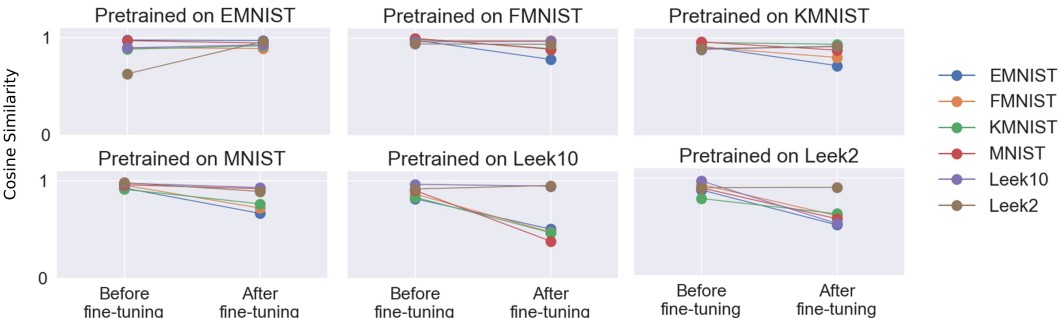

Figure 10: Cosine similarity analysis before and after 1-location fine-tuning, for each pretraining fully-translated dataset, showing that in almost every condition the network acquired translation invariance, but it was some times disrupted by fine-tuning.

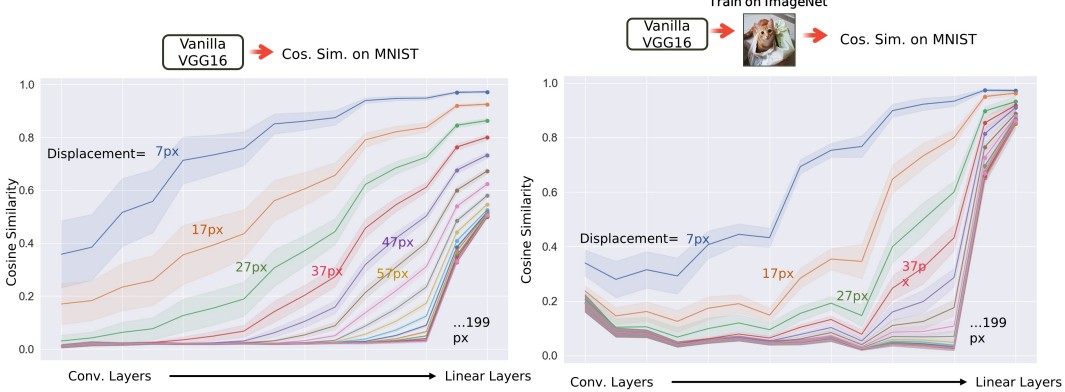

Figure 11: Cosine Similarity value across the whole network, for several displacements, for the vanilla network and a network pretrained on ImageNet.

### A.3.2 CONVOLUTION AND TRANSLATION INVARIANCE

We have argued in the Discussion that the convolution operation is not responsible for coding the property of translation invariance. Theoretically, this is due to convolution being equivariant rather than invariant. Practically, we can measure this by computing the cosine similarity across both the convolutional layers and the fully connected layers (whereas the cosine similarity in Section 3.6 was performed on the penultimate layer, so a fully-connected one). We ran the analysis on a vanilla network and a network pretrained on ImageNet (Figure 11). This plot shows us three things: 1) as expected, as displacement increases, the convolutional layers encode less and less of the similarity between the translated images. For the pretrained network, after 37 pixels, most of the similarity seems to be captured uniquely by the fully connected part of networks. 2) As seen in Section 3.6, there seems to be a very small amount of translation tolerance even for a vanilla network. The similarity is high for translations until about 27 pixels. This may be due to some relationship with pooling, convolution, and the size of the items used (50 pixels). 3) Cosine Similarity on the

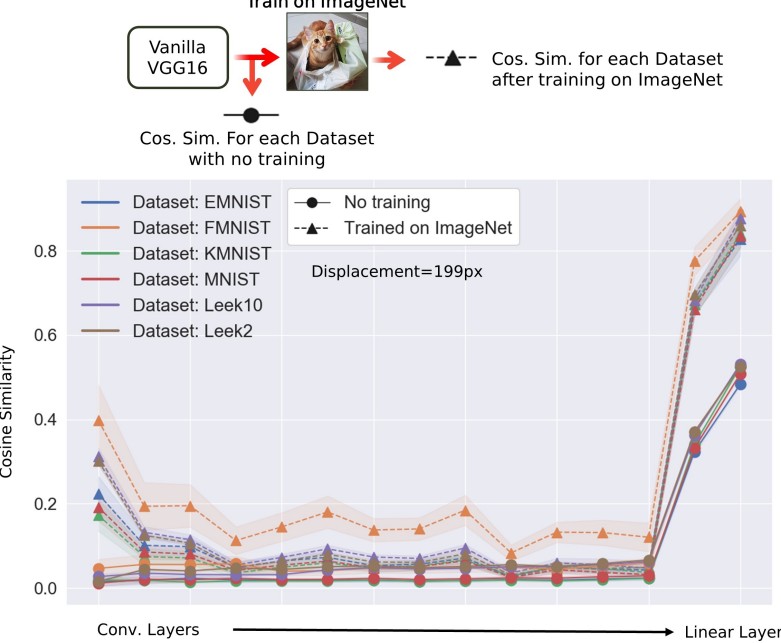

Figure 12: Cosine Similarity values across the whole network, at the most extreme displacement, for the vanilla network and a network pretrained on ImageNet.

convolutional layers seem much more pronounced on the vanilla than on the pretrained network. It seems that pretraining makes the convolution *less* involved in translation invariance, pushing the encoding onto the fully-connected layer. Overall, we noticed that the convolutional layers may be responsible for a limited degree of translation invariance. However, for larger displacement, the translation invariance seem to be completely encoded into the fully connected layers, as shown in Figure 12 with the maximum displacement of 199 pixels.

### A.4 Training a Fully Connected Network

If, as argued in the Discussion and in Section A.3.2, convolutions are not responsible (or minimally responsible) for translation invariance, it should be possible to train a fully connected network to acquire translation invariance. We chose the architecture tested in Cireşan et al. (2010), which proved able to obtain 0.35% test error on the MNIST dataset. The architecture has 4 hidden layers of $(2500, 2000, 1500, 1000, 500)$ units, and ReLu was applied after each layer. We pretrained the network with a fully-translated dataset: we used MNIST and Leek10 (without any data augmentation). We trained until convergence (corresponding to $\sim$99% accuracy). We then fine-tuned the network on 1-location datasets and tested on a fully-translated version of that dataset. The network pretrained on MNIST was fine-tuned and tested on Leek10 and Leek2. The network pretrained on Leek10 was fine-tuned and tested on MNIST and Leek2. We ran the experiment 5 times for each condition.

The results are shown in Figure 13. We observed that a purely fully-connected network only showed translation invariance when pretrained on Leek10 and fine-tuned on Leek2. The lack of invariance when pretrained on Leek2 or 10 and fine-tuned on MNIST can be explained by catastrophic interference (Section 3.6) due to fine-tuning on a more complex dataset than the pretrained one. However the lack of invariance when pretraining MNIST and fine-tuning on Leek10 is puzzling. It is possible that a fully-connected network is much more prone to catastrophic interference compared to a CNN due to lack of weight sharing. Therefore, even though it may be possible for a fully-connected network to learn translation invariance, and even though the convolutional layers are not directly responsible for this ability, the convolution operation is clearly a major facilitator in transferring this ability to a new dataset.

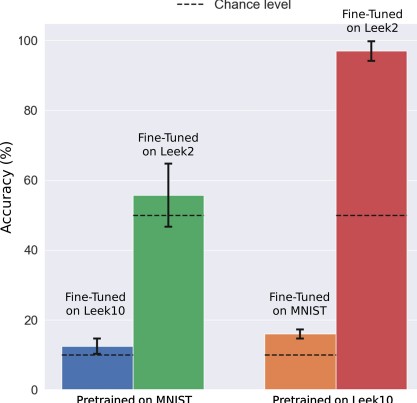

Figure 13: Accuracy of a fully-connected network pretrained on fully-translated MNIST or Leek10, and fine-tuned on 1-location MNIST, Leek10, or Leek2. The network pretrained on Leek10 was able to exhibit translation invariance for Leek2, but the other combinations failed, probably due to catastrophic interference being more problematic for a non-convolutional network.

