# OpenReview forum: "Convolutional Neural Networks are not invariant to translation, but they can learn to be"
_ICLR.cc/2021/Conference — Reject_

### Official Review · AnonReviewer2 · 2020-10-29
**Review on Convolutional Neural Networks are not invariant to translation, but they can learn to be**

**Rating:** 5
**Confidence:** 5

**Review:**

This paper analysis and studies translation invariance in convolution neural networks. It argues that typically it is claimed that CNNs are translation invariant due to the convolution function, and that actually convolution are equivariant. While pooling is the actual function that gives local invariance (or global when the pooling is across all locations), this is not included in the description of invariance. One neural network, VGG-16, is used for the analysis in different scenarios: 1) pre-trained on Imagenet and fine-tuned to the new dataset on one location 2) trained from scratch using the new dataset in one location 3) trained from scratch using the new datasets in all locations of the canvas and test on the other datasets. The main conclusion in the paper is that CNNs are not invariant to translation by design of the architecture, but that when pre-trained on naturalistic images, they can be.


Positive points:

- the research question is important and this kind of analysis to understand CNNs are crucial to better understand network capabilities and being able to predict their behaviour.

----
COMMENT AFTER REBUTTAL PERIOD:
Given that there was no rebuttal, I keep my initial rating.

- study how using pre-trained networks affect the generalization in some factors, e.g. translation invariance in this paper, is interesting and novel.

- The paper is well written and easy to follow.



Concerns:

- The main concern is about the experimental set-up and results presented in the paper. Only one network is used for the study, to validate if it is indeed the architectural design the gives translation invariance. It is a weak statement if other CNNs architectures with different configurations (number of layers, amount of local pooling, if global pooling is used, the effect of strides, etc ) analyzed, and the current observations might only apply to VGG-16.

- The results in figure 2 B when comparing the translation invariance across the different datasets, it is mentioned that depending on the dataset that the network is initialized from it brings more or less translation invariance. A deeper analysis on the amount of data used, the similarity between the objects across datasets, and how this effect the final performance would be nice to include. Here talking about invariance is a bit confusing, and it might be that this robustness to position changes is more related the generalization properties to the other datasets due to the resemblance of the objects, and obtained by experience.

- It would have been nice to see the behaviour in another transformation as well, since it would strengthen the claim that the invariance (I would say robustness) to transformations is not due to the architecture, but to previous exposure to naturalistic images, and if it is not, it would bring some light into why it is for translation and not for other transformations.




Minor comments:

* As a side comment, since there is some motivation in the introduction relating to human visual processing, humans have a loss of acuity (recognition performance) with distance to the focal point, and perceive high accuracy in the fovea a low-accuracy in the periphery, which is not captured by typical CNNs.

---

### Official Review · AnonReviewer4 · 2020-10-29
**Interesting experimental setup to evaluate translation invariance, but insights are not significantly new**

**Rating:** 4
**Confidence:** 3

**Review:**

This paper addresses the problem of how convolutional neural networks (CNNs) achieve translation invariance, and the authors argue that this invariance es mostly learned from suitable datasets, rather than a result of the architecture. In particular, ImageNet-pretrained networks have learned to be invariant to translation, and fine tuned. The experiments are performed in MNIST-like datasets evaluating classification performance at different locations. The authors conclude that invariance is achieved when the CNN is trained with the different objects being presented at different locations across the canvas, and that the invariance can be forgotten after subsequent training.

Pros:
- The experiments explore several settings and are convincing in clearly showing that translation invariance requires that the network observes the different objects translated across the canvas, in their particular setting (although I have some concerns about the setting).
- Understanding how neural networks achieve certain properties, such as translation invariance, is very relevant.
- The paper is well written and can be followed easily. I like particularly the ilustrations of the experiments.


Cons:
- The main concern I have is that the insights are relatively incremental. The experimental setting replicates Blything et al 2020 and the main insight of CNNs can learn translation invariance from suitable large and diverse datasets such as ImageNet was already shown in that paper (note that, although available in arxiv in Sept 2020, the authors are aware of the work, since they state that they use Bything et al.'s dataset and replicate their experiments). The results in the submission are not showing significantly novel insights.
- Results are shown with small datasets (MNIST-like), but not clear how they extrapolate to more complex one. It is also not clear to me that is possible to train a heavy model such as VGG16 with such small resolution datasets, even when they are translated to different locations. It probably results in very significant overfitting.
- Only evaluated on VGG16. To be more convincing in the general claim, it is necessary to also evaluate other models.
- The authors only analyze translation invariance of the whole network. It would be more interesting to analyze invariance of the different layers via intermediate representations. Experiment 3 for instance, encourages local translation invariance (within each quadrant), but not across the whole canvas. I would expect that higher layers still behave like Vanilla VGG16 which overfits to the location, while lower layers show higher level of invariance.
- Fine tuning the whole network (I understand the authors train/fine tune all the layers) in this setting is probably leading to significant overfitting and therefore to catastrophic forgetting. The authors should consider the case where only the classifier is trained (and a variable number of layers in the top), and thus less prone to overfitting and avoid forgetting in lower layers, to further assess the invariance in different layers.

Questions
Please clarify cons.

Minor comments
Some figures seem to suggest that images have multiple objects in multiple locations. My understanding is that every image has only one object, and the location can change, so those figures may be misleading. Please clarify and modify the figure if necessary.

---

### Official Review · AnonReviewer3 · 2020-11-01
**through experiments but lacks novelty**

**Rating:** 4
**Confidence:** 4

**Review:**

This paper examines the source of translational invariance in CNNs and points out that the convolution operation is translationally equi-variant and not invariant. Authors thoroughly examine how training and architecture contribute to translational invariance in CNNs. My main concern about this paper is with respect to its novelty. While the experiments are done in a thorough way, the main point of this paper (that translation invariance is formed during training) is mostly known to the community. The equi-variance property of convolution operation is discussed in machine/deep learning textbooks (e.g. [2]) and the importance of image-augmentations like spatial jitter on network object recognition performance has repeatedly been demonstrated. Because of this reason I don’t think this paper would be suitable for publication at ICLR.

Other comments:
* The contribution of the paper is very unclear from the abstract, even getting past the first two sections I was still left wondering what I should be expecting to see in the rest of the paper.
* The text needs more proofing reading, more than few typos and misuse of words. E.g. bases —> basis; human vision recognition —> human visual (object) recognition
* LeCun et al. 1998 is cited as a biological model which is not a good example of a biological neural network. Although CNNs have some commonalities with biological neural networks they have many more differences. [1] might be a better reference to an early biologically inspired neural net
* It is claimed that “CNNs achieve neither rotation nor scale invariance”. However this is not a binary property. What matters is the degree of invariance that could be measured.
* section 3: by "non-pretrained network” do you mean the untrained network?
* It is unclear from the text under section 3 what “pretraining on the whole canvas” is
* the 1-location dataset is not described anywhere in the paper and I had to go by a guess as to what this dataset contains.
* in section 5, the use of the cosine similarity measure instead of the accuracy which was used in the previous 4 experiments is not well motivated. If this is a better measure why not using it in all experiments.


[1] Fukushima, K. (1975). Cognitron: A self-organizing multilayered neural network. Biological cybernetics, 20(3-4), 121-136.

[2] Goodfellow, I., Bengio, Y., Courville, A., & Bengio, Y. (2016). Deep learning (Vol. 1, p. 2). Cambridge: MIT press.

---

### Decision · Program_Chairs · 2021-01-07
**Final Decision**

**Decision:**

Reject

**Comment:**

This paper receives 3 initial rejection ratings. No rebuttal was submitted by the authors. There is no basis for overturning the reviewers' decisions. This paper should be rejected.